# Multivariate mixed-effects ordinal logistic regression models with difference-in-differences estimator of the impact of WORTH Yetu on household hunger and socioeconomic status among OVC caregivers in Tanzania

Amon Exavery[1,2]*, Peter J. Kirigiti[1], Ramkumar T. Balan[1], John Charles[2]

**1** Department of Mathematics and Statistics, College of Natural and Mathematical Sciences (CNMS), The University of Dodoma, Dodoma, Tanzania, **2** Pact Tanzania, Dar es Salaam, Tanzania

\* aexavery@gmail.com

## Abstract

### Background

Although most of the livelihood programmes target women, those that involve women and men have been evaluated as though men and women were a single homogenous population, with a mere inclusion of gender as an explanatory variable. This study evaluated the impact of WORTH Yetu (an economic empowerment intervention to improve livelihood outcomes) on household hunger, and household socioeconomic status (SES) among caregivers (both women and men) of orphaned and vulnerable children (OVC) in Tanzania. The study hypothesized that women and men respond to livelihood interventions differently, hence a need for gender-disaggregated impact evaluation of such interventions.

### Methods

This is a secondary analysis of longitudinal data, involving caregivers' baseline (2016–2019) and follow-up (2019–2020) data from the USAID Kizazi Kipya project in 25 regions of Tanzania. Two dependent variables (ie, outcomes) were assessed; household hunger which was measured using the Household Hunger Scale (HHS), and Socioeconomic Status (SES) using the Principal Component Analysis (PCA). WORTH Yetu, a livelihood intervention implemented by the USAID Kizazi Kipya project was the main independent variable whose impact on the two outcomes was evaluated using multivariate analysis with a multilevel mixed-effects, ordinal logistic regression model with difference-in-differences (DiD) estimator for impact estimation.

**Data Availability Statement:** All relevant data are within the manuscript and its Supporting Information files.

**Funding:** The authors received no specific funding for this work.

**Competing interests:** The authors have declared that no competing interests exist.

**Abbreviations:** aOR, adjusted Odds Ratio; GII, Gender Inequality Index; HHS, Household Hunger Scale; HIV, human immunodeficiency virus; LHIV, living with human immunodeficiency virus; MRCC, Medical Research Coordinating Committee; OVC, orphans and vulnerable children; PCA, Principal Component Analysis; SES, socioeconomic status; UNDP, United Nations Development Programme; USAID, United States Agency for International Development.

## Results

The analysis was based on 497,293 observations from 249,655 caregivers of OVC at baseline, and 247,638 of them at the follow-up survey. In both surveys, 70% were women and 30% were men. Their mean age was 49.3 (±14.5) years at baseline and 52.7 (±14.8) years at the follow-up survey. Caregivers' membership in WORTH Yetu was 10.1% at the follow-up. After adjusting for important confounders there was a significant decline in the severity of household hunger by 46.4% among WORTH Yetu members at the follow-up compared to the situation at the baseline (adjusted Odds Ratio (aOR) = 0.536, 95% Confidence Interval (CI) [0.521, 0.553]). The decline was 45.7% among women (aOR = 0.543 [0.524, 0.563]) and 47.5% among men (aOR = 0.525 [0.497, 0.556]). Regarding SES, WORTH Yetu members were 15.9% more likely to be in higher wealth quintiles at the follow-up compared to the situation at the baseline (aOR = 1.159 [1.128, 1.190]). This impact was 20.8% among women (aOR = 1.208 [1.170, 1.247]) and 4.6% among men (aOR = 1.046 [0.995, 1.101]).

## Conclusion

WORTH Yetu was associated with a significant reduction in household hunger, and a significant increase in household SES among OVC caregivers in Tanzania within an average follow-up period of 1.6 years. The estimated impacts differed significantly by gender, suggesting that women and men responded to the WORTH Yetu intervention differently. This implied that the design, delivery, and evaluation of such programmes should happen in a gender responsive manner, recognising that women and men are not the same with respect to the programmes.

## Introduction

Evaluation of programme impacts remains methodologically complex, especially in non-experimental settings [1–3]. Experimental designs with randomisation or randomised controlled trials (RCTs) are universally accepted as the gold standard for gauging programme impacts [1, 3]. This unique feature stems from their inherent ability to achieve similarity between treatment and control subjects in terms of both measured and unmeasured characteristics [4–7]. However, RCTs are expensive to conduct and often involve many ethical issues, hence limited applicability [8]. Due to this, non-experimental designs such as longitudinal, cohort, or case-control have been recommended as methodological alternatives [9–12]. However, in non-experimental settings, the similarity of subjects cannot be guaranteed due to lack of randomisation, and being in treatment or control groups occurs on a self-selection basis, resulting into selection bias as a major limitation [1, 3]. Therefore, because of the selection bias, estimated impacts from non-experimental programmes can be biased by unmeasured confounders [13–15]. In this case, impact evaluation calls for methodological approaches that can minimise selection bias as much as possible by accounting for as many variables (ie, sources of bias) as possible in the regression models or matching to reduce the variance represented by the error term and increase the precision of the programme impact [16]. Further, impact evaluation of non-experimental programmes in different fields may require field-specific approaches to further minimise selection bias and improve the precision of the estimated impacts.

In the area of economic empowerment programmes, a massive evolution has happened over the last few decades with diverse programming modalities, hence the complex evaluation of their impacts [17]. Extant evidence reveal clearly that, most of such programmes target women [18–20]. Core explanations for this include the fact that in many parts of the world, women and girls suffer a substantial share of various forms of discrimination and vulnerability [21]. Also, while women are the ones mostly affected by poverty, they are the key players in food production for their families, especially in low- and middle-income countries (LMICs) [22, 23]. Another explanation is that economic empowerment programmes to enhance livelihood outcomes are delivered as part of structural interventions for the prevention of Human Immunodeficiency Virus (HIV) because women are at higher risk for HIV acquisition than men [19]. According to the United Nations Development Programme (UNDP), Tanzania's Gender Inequality Index (GII) for the year 2021 stood at 0.560, placing the country at the 146th position globally [24]. With this GII which surpassed the world's average of 0.465, Tanzania was classified as having low human development, underscoring the persistent challenge of inequality between women and men in the country [24]. Therefore, recognising that gender inequality propels inequities in multiple dimensions including health and wellbeing [25], the United Nations fifth goal for sustainable development was explicitly set to achieve gender equality and empowerment of all women and girls [26]. Due to this, several empowerment programmes have been implemented in the world, including those addressing food insecurity, hunger, and poverty [27, 28].

Considering this background, evidence suggests that in economic empowerment programmes where both genders are involved (eg, Kakuhikire et al. [29]), it is likely that women and men respond to the interventions differently. Unfortunately, previous impact evaluations of the programmes involving both genders have treated women and men as a single homogenous population, lacking gender-disaggregation of the impacts. Attempts to integrate gender in some evaluations have mainly elucidated the interplay between gender norms and livelihood programming [30, 31], comparing women and men in terms of livelihood activities [19, 32], and earnings [19]. Despite this, impact evaluation of gender-responsive programmes has not been so common. Yet it is equally needed to inform strategies for efficient design, delivery, and evaluation of livelihood interventions for gender-equitable livelihood outcomes.

In view of this, this paper evaluates the impact of WORTH Yetu on livelihood outcomes, namely, household hunger, and household socioeconomic status (SES). The study also attempts to recognise the differences between women and men in terms of the impact of WORTH Yetu on each of the two outcomes as a way of being scientifically informed regarding the significance of gender disaggregation in impact evaluation of non-experimental economic empowerment programmes. The analysis herein is based on WORTH Yetu, an economic empowerment intervention implemented under the USAID Kizazi Kipya project (2016–2021) to improve livelihood outcomes among caregivers of orphans and vulnerable children (OVC) in Tanzania [33]. The project was geographically large-scale and nationally representative, reaching hundreds of thousands of households in Tanzania caring for OVC to improve their health and wellbeing. The analysis of WORTH Yetu impact on livelihood outcomes (ie, household hunger, and socioeconomic status (SES)) is intended to go beyond the common computation of frequencies and percentages by gender or mere inclusion of gender as a control variable in regression analysis, (eg, Embleton et al. [34]) but advances to handling men and women as separate populations and proceeding to evaluate programme impacts in each, then comparing, contrasting, and explaining their similarities, differences, and implications. This will generate evidence of not only how effective the intervention was, but also inform options and strategies for achieving more effectiveness of the programmes, thereby contributing to the core purposes of impact evaluation–accountability and learning [2].

This evaluation approach is consistent with the Realist Evaluation (RE) theory [35]. The RE theory was designed to enhance the understanding of how different interventions work in different contexts. The theory attempts to elucidate what works, for whom, in which situations, why does it or does not work and over what duration [36, 37]. As such, the theory assumes that no intervention works everywhere for everyone, which is why context is vital. The theory deals with the causal mechanism that enables the programme to work [37]. The theory operates on the configurations of C + M = O, whereby C represents the context, M stands for the mechanism and O represents the outcome. The configuration describes how specific contextual factors work to trigger mechanisms, and how the combination brings about certain outcomes [35].

## Materials and methods

### Study design and settings

The present study is longitudinal in design, involving secondary data from the USAID Kizazi Kipya project. The data are from 81 district councils in 25 regions of Tanzania where the project enrolled beneficiaries (baseline) during 2016–2019, with a follow-up assessment during 2019–2020. Regions in Tanzania where the USAID Kizazi Kipya project was not implemented were excluded from this study as the necessary data were unavailable. The project aimed to scale-up the uptake of HIV services, other health services, as well as social services by Tanzanian OVC and their caregivers through a Pact-led consortium of non-governmental organizations (NGOs), Civil Society Organizations (CSOs), and the Government of the United Republic of Tanzania at national, regional, district, and community levels [38]. The project provided services in the areas of health, economic empowerment, education and other social services to OVC, vulnerable youth and their caregivers. At the community level, volunteers known as Community Case Workers (CCWs) and Lead Case Workers (LCWs) supported the implementation of the project by identifying services needed by each enrolled beneficiary, then proceeding to delivery of certain services while providing referrals for other services that the CCWs and LCWs were not mandated to provide directly.

### Study area

The 25 regions represented by the data analyzed herein for the present study are Arusha, Dodoma, Dar es Salaam, Geita, Iringa, Kagera, Katavi, Kigoma, Kilimanjaro, Mara, Mbeya, Mjini Magharibi, Morogoro, Mtwara, Mwanza, Njombe, Pwani, Rukwa, Ruvuma, Shinyanga, Singida, Simiyu, Songwe, Tabora, and Tanga.

### Study population

OVC caregivers aged 18 years or more constituted the current study population. The caregivers included herein were the beneficiaries of the USAID Kizazi Kipya project enrolled (ie, baseline) from 24th November 2016 to 30th October 2019 and later reassessed in a followed-up survey from 1st February 2019 to 30 September 2020. From enrollment to the follow-up survey, each caregiver was assessed twice with the FCAA tool. By definition, the project defined a caregiver as a parent/guardian who has the greatest responsibility (ie, primary) in caring for one or more OVC in one household [39]. In the enrollment process, one primary caregiver in a household was registered in the project and issued a unique caregiver identification number (CGID). Therefore, the number of caregivers was equal to the number of households enrolled in the project.

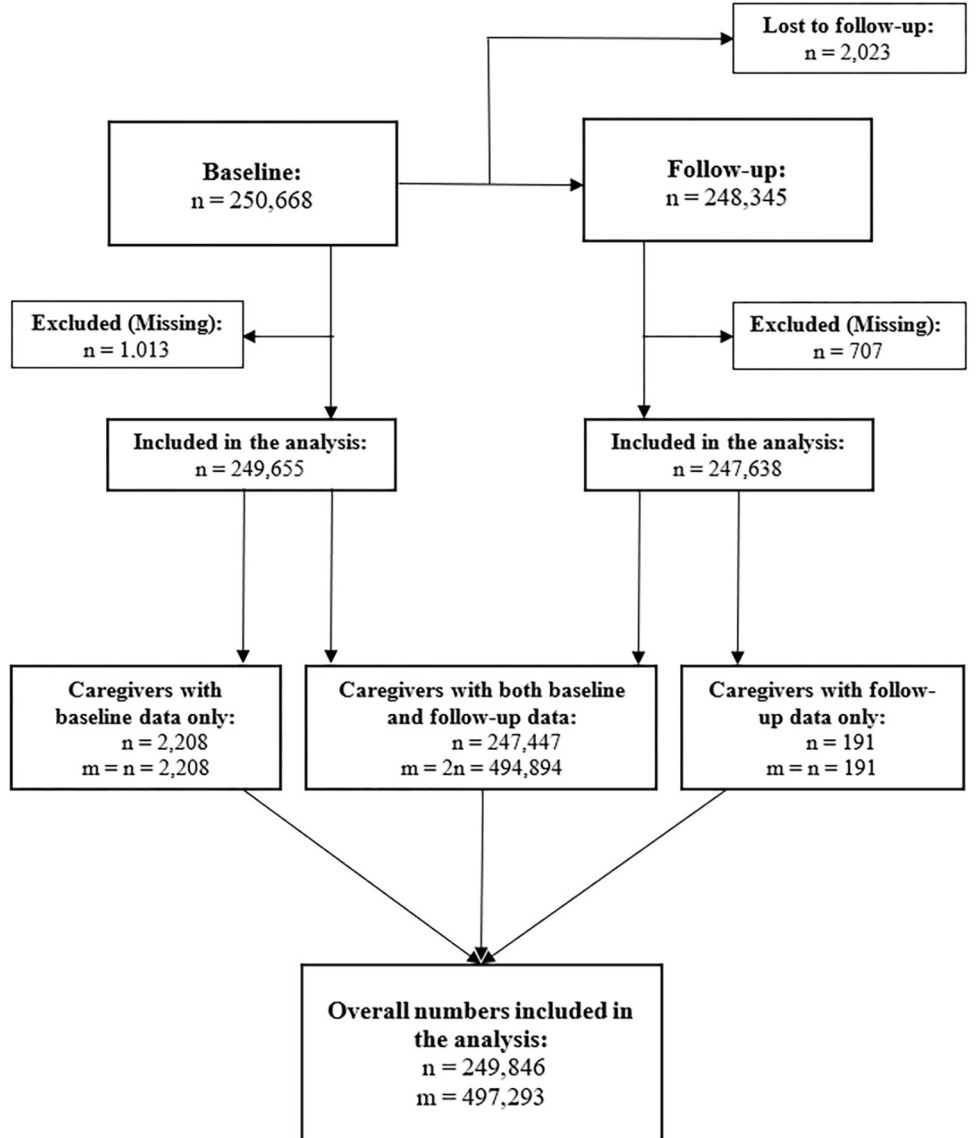

**Fig 1. Flow diagram of the number of OVC caregivers (n) and their observations (m) included in the analysis at baseline and follow-up.** Notes for Fig 1 n represents the number of caregivers, and m represents the number of caregivers' observations. m = 2n: Each caregiver has two observations; one at the baseline and another at the follow-up. m = n: Each caregiver has one observation only; either at the baseline or at the follow up.

Fig 1 is the flow diagram of the number of caregivers and their baseline and follow-up observations included in the analysis. In the extraction process, 250,668 caregivers had matching CGID in the baseline and follow-up datasets from the USAID Kizazi Kipya project database. Although 2,323 of these had matching CGIDs in the follow-up dataset, they had no data on all the variables, suggesting that they were lost to follow-up (LTFU), leaving 248,345 caregivers at baseline with data at follow-up. Upon further explorations of the datasets, 1,013 and 707 caregivers from baseline and follow-up datasets, respectively, were excluded because they had missing observations in one or more of the variables included in the analysis. This process resulted in 249,655 caregivers at baseline and 247,638 caregivers at the follow-up with eligible data for the current analysis. Since this was a longitudinal study, with each caregiver expected

to have two observations (ie, one at baseline and another one at follow-up), the analysed data was distributed as 247,447 caregivers each observed twice, making up 247,447*2 = 494,894 observations; 2,208 caregivers with baseline data only, making up 2,208*1 = 2,208 observations; and 191 caregivers with follow-up data only, making up 191*1 = 191 observations. Therefore, the final analysis was based on 249,846 caregivers with a total of 497,293 observations representing baseline and follow-up assessments (Fig 1).

## Data source

As highlighted, the USAID Kizazi Kipya was a community-based project implemented in Tanzania for five years, from 2016 to 2021. The primary beneficiaries of the project were OVC and their caregivers. The project provided services to enrolled beneficiaries in several dimensions, including HIV services, other health services, food and nutrition, psychosocial care and support, and economic strengthening (ES).

## The WORTH Yetu intervention

From the project, data pertaining to the ES intervention was sought for the purpose of the present study. The ES intervention under the project was intended to ensure that the caregivers have the financial resources to meet the needs of the OVC by improving their livelihoods, employment skills, and life skills as a critical pathway towards growth and reduction of their economic vulnerability. All caregivers were eligible, and hence informed of the ES intervention under the project, but membership or participation in the intervention was voluntary. The ES intervention was delivered through WORTH Yetu groups which were locally formed, requiring members to meet weekly with mandatory and voluntary savings to create a base for individual loans as well as startup projects for the groups. WORTH Yetu members had access to financial literacy, an opportunity to save as well as access to microcredits from financial institutions and other sources. Through the WORTH Yetu groups, members were enabled to start group projects, such as farming, animal husbandry, and horticulture [40].

The WORTH Yetu groups were facilitated by Livelihoods Volunteers (LVs), a cadre formerly recruited at the ward level with support from the respective Ward Executive Officers (WEOs). Their recruitment process involved written and oral interviews as well as vetting by the Local Government Authority (LGA) at the ward level. Each LV supported a maximum of 10 groups with at least 150 members from targeted OVC households. The USAID Kizazi Kipya project provided financial and technical inputs to support LVs to deliver quality services to the WORTH Yetu groups. This required each LV to attend training sessions organized by CSO's Economic Strengthening and Livelihood Officers (ESLOs) on key curriculums for the effective functioning of WORTH Yetu groups. The trained LV cascaded the training to the management committee of each WORTH Yetu group and members. Each group received two or more visits every month from the LVs. The LV was also responsible for coaching groups and facilitating linkages to other ES opportunities. It was also a requirement that the LV meets their respective ESLO every month for training and submission of progress reports of their groups.

## Variables

**Dependent variables.** Two dependent variables (ie, outcomes) were assessed in the present study, namely, the level of household hunger, and household socioeconomic status (SES). Both variables were measured objectively as ordinal variables as described below.

*Household hunger*. The level of household hunger was determined using the Household Hunger Scale (HHS). The HHS was established by the Food and Agriculture Organization

(FAO) and the Tufts University through the Food and Nutrition Technical Assistance III Project (FANTA) [41]. The HHS is an improved version of the Household Food Insecurity Access Scale (HFIAS). The HHS was formed by reducing the HFIAS to three questions after internal and external validations in Africa and Asia [41, 42]. The three questions which the HHS utilises in the determination of the level of household hunger are: (1) "In the past 4 weeks, how often was there ever no food to eat of any kind in your household because of lack of resources to get food?", (2) "In the past 4 weeks, how often did any household member go to sleep at night hungry because there was no enough food?", and (3) "In the past 4 weeks, how often did any household member go whole day and night without eating anything?". Each of the three questions has four possible responses: never, rarely (once or twice), sometimes (3 to 10 times) and often (more than 10 times). According to the HHS, these responses are reorganised in such a way that the first category, never, is coded '0', the subsequent two categories, rarely (once or twice) and sometimes (3 to 10 times) are coded '1', and the last category, often (more than 10 times), is coded '2'. Then a row sum of the codes across the three questions is computed to generate a hunger score for each household. The resulting variable has hunger score values ranging from 0 to 6, such that the higher the score the more severe the level of household hunger. Ultimately, based on the scores, the HHS classifies households in three hunger-defining categories in the increasing order of hunger severity as (1) little to no hunger, (2) moderate hunger, and (3) severe hunger [41] as mathematically represented below.

$$\text{Level of household hunger} = \begin{cases} 1, & \text{Little to no hunger, if hunger score} = 0,\ 1 \\ 2, & \text{Moderate hunger, if hunger score} = 2,\ 3 \\ 3, & \text{Severe Hunger, if hunger score} = 4,\ 5,\ 6 \end{cases}$$

*SES*. Household SES was assessed using the Principal Component Analysis (PCA) of household-owned assets [43]. Household assets involved in the PCA were whether the main dwelling material for the household is concrete, cement, aluminium and/or other materials, whether the household owns land, chicken, goats, cows, bicycles, tractors, motorcycles, motor vehicles, ovens, and hair driers. The final SES variable from the PCA was ordinal in structure, with five categories known as wealth quintiles, from the lowest quintile (Q1) to the highest quintile (Q5) for the poorest households and the wealthiest households, respectively, as represented below.

$$\text{Socioeconomic status (SES)} = \begin{cases} 1, & \text{Lowest (Q1)} \\ 2, & \text{Second (Q2)} \\ 3, & \text{Middle (Q3)} \\ 4, & \text{Fourth (Q4)} \\ 5, & \text{Richest (Q5)} \end{cases}$$

**Independent variables.** WORTH Yetu was the main independent variable of interest for this study. This was the livelihoods intervention programme whose impact on the dependent variables described above was assessed. The variable was binary, representing whether the caregiver was a member ('1') or not a member ('0') of WORTH Yetu between baseline and follow-up periods of assessment. Gender was another main independent variable which was time-independent, recognizing the caregiver as either a man or a woman based on their self-identification during enrollment.

Other independent variables included were age in groups of 18–29 years, 30–39 years, 40–49 years, 50–59 years, and 60+ years. Level of formal education attained was also included amongst the independent variables (never attended, primary, and secondary or more), as well as marital status (married or living together, divorced or separated, never married, and widow or widower), family size (2–3 people, 4–6 people, and 7+ people), whether one or more family members has health insurance (yes, and no), HIV status (negative, positive, and unknown or undisclosed), place of residence (rural and urban), and whether the caregiver was physically or mentally disabled (yes, and no). The disability was assessed at enrollment based on physically observable conditions and limitations of the caregiver, such as blindness, physical disability etc. as described elsewhere [44]. The source of data and all variables used for this study is the baseline and follow-up surveys conducted among OVC caregivers of the USAID Kizazi Kipya project in Tanzania.

## Data analysis

Both descriptive and inferential statistical techniques were applied in the current study. In the descriptive part, the frequency distribution of the respondents was computed through one-way tabulations of each of the variables at baseline and follow-up. This was followed by two-way tabulations of each of the outcomes by WORTH Yetu and each of the described independent variables, with a Chi-square ($\chi^2$) test to gauge the degree of association between them.

In the inferential analysis, multivariate analysis to evaluate the impact of WORTH Yetu on both household hunger, and SES was conducted using a multilevel mixed-effects ordinal logistic regression model with a DiD estimator using Stata's "meologit" syntax. The model operates on the condition that numerical values representing the categories of each of the outcomes are not relevant, except that larger values correspond to higher outcomes. The choice of this model was motivated by a consideration that both outcomes were inherently ordinal variables, with the underlying assumption being that the three categories of household hunger are in the increasing order of hunger severity, and the five categories of SES (ie, wealth quintiles) are in the order of increasing socioeconomic wellbeing. In both cases, the categories have natural ordering, but the distances between adjacent categories of each variable are unknown [45].

Also, since the data was longitudinal, we assumed that observations of the same caregiver are correlated. So, a two-level multilevel model with a random intercept was defined, whereby observations (level 1) were nested within caregivers (level 2). In the analysis, a full model was fitted for all observations of the caregivers, after which separate models–one for women's observations and another for men's observations–were then fitted. In both cases, WORTH Yetu impact on each of the outcomes was evaluated using the DiD estimator through an interaction term between WORTH Yetu ('0' = non-member, and '1' = member) and time ('0' = baseline, and '1' = follow-up), controlling for potential confounders. The full model was used to gauge the overall WORTH Yetu impact on each of the outcomes, as well as the significance of gender as an indicator of how similar or different women and men responded to the WORTH Yetu intervention. The purpose of the separate models (for women and men) was two-fold: (1) to compare the magnitude of the impact of WORTH Yetu on household hunger and SES between men and women, and (2) to compare the extent to which other factors that influence household hunger and SES are similar or different between men and women.

The basic form of a two-level multilevel model for an ordinal outcome variable with a random intercept can be described as follows. Given an ordinal outcome variable such as SES, the basic conception is that behind the observed ordinal variable, there exists an underlying latent continuous variable that is not measured directly [46]. Denoted as $Y_{ij}^*$, a model for the latent

continuous outcome variable can be represented as follows, considering the context of the current study: -

$$Y_{ij}^* = \beta_1 P_{ij} + \beta_2 T_j + \beta_3 (P_{ij} * T_j) + \beta_4 X_{ij} + u_{0j} + \varepsilon_{ij}$$

To link the $Y_{ij}^*$ and the observed ordinal outcomes $Y_{ij}$, a threshold model is defined. For $Y_{ij}$ ordinal categories, $c = 1, 2, 3, \ldots, C$, a threshold model can be represented as

$$Y_{ij} = \begin{cases} 1, & \text{if } Y_{ij}^* \leq k_1 \\ 2, & \text{if if } k_1 < Y_{ij}^* \leq k_2 \\ 3, & \text{if } k_2 < Y_{ij}^* \leq k_3 \\ \vdots \\ C, & \text{if } Y_{ij}^* > k_{c-1} \end{cases}$$

Where: $k_c$ is a threshold parameter, and the thresholds are in an increasing order, such that $k_1 < k_2 < k_3 \ldots < k_{c-1}$. When $Y_{ij}^*$ increases past a given threshold, there is a discrete jump in the observed ordinal/ordered categories of $Y_{ij}$. For example, when $Y_{ij}^*$ exceeds the threshold $k_1$, $Y_{ij}$ changes from 1 to 2; when $Y_{ij}^*$ exceeds the threshold $k_2$, $Y_{ij}$ changes from 2 to 3, etc.

The random effects at level 2 are assumed to be normally distributed, such that, $u_{oj} \sim N(0, \delta_u^2)$ for all caregivers. For level 1 residuals, and considering the logit specification, $\varepsilon_{ij} \sim \text{logistic}(0, \pi^2/3)$ for all observations, leading to a multilevel cumulative logit model as described by Bauer and Sterba (2011) [46]. The random parameters are independent of one another–ie, $\text{Cov}(u_{oj}, \varepsilon_{ij}) = E(u_{oj}) = E(\varepsilon_{ij}) = 0$

In the framework of generalized linear models, the same cumulative multilevel logit model is expressed as:

$$\Pr(Y_{ij}) = f^{-1}\left[k_c - \eta_{ij}\right] = \frac{1}{1 + e^{-(k_c - \eta_{ij})}} = \frac{1}{1 + e^{-(k_c - [P_{ij} + \beta_2 T_j + \beta_3 (P_{ij} * T_j) + \beta_4 X_{ij} + u_{0j}])}}$$

Where $\Pr(Y_{ij})$ is the cumulative probability that a response of the ordinal outcome variable will be recorded in category k or below. $\eta_{ij} = \beta_1 P_{ij} + \beta_2 T_j + \beta_3 (P_{ij} * T_j) + \beta_4 X_{ij} + u_{0j}$ is a linear predictor constituting a linear combination of WORTH Yetu and other observed factors and random effects. Again, $k_c$ is a threshold parameter, $f^{-1}(.)$ is the inverse link function that maps the continuous nature of $[k_c - \eta_{ij}]$ into the asymptotes of 0 and 1 of the predicted values [47, 48].

## Ethics approval and consent to participate

This study received an ethics approval from the Institutional Research Review Ethics Committee (IRREC) of the University of Dodoma in Tanzania (MA.84/261/61/57). The data had a prior ethics approval from the Medical Research Coordinating Committee (MRCC) of the National Institute for Medical Research (NIMR) in Tanzania, with ethics clearance certificate number NIMR/HQ/R.8a/Vol.IX/3024, also described elsewhere [39]. The data represent beneficiaries of the USAID Kizazi Kipya project whose households were enrolled in the project voluntarily. The screening and enrollment form included a section where caregivers who consented to participate in the project signed as evidence that they had been informed about the project, and that they were voluntarily willing to participate. Datasets provided for this study were anonymous, securely stored, and only accessible to the authors.

# Results

## Profile of respondents

The present analysis was based on observations from 249,655 caregivers at the baseline, and 247,638 of them at the follow-up survey. By gender, 70.0% of the caregivers were women and the rest 30.0% were men. Overall, their mean age was 49.3 (±14.5) years at baseline and 52.7 (±14.8) years at the follow-up survey. These values were different by gender as women were relatively younger than men. At the baseline, women's mean age was 48.0 (±14.4) years and men's was 52.3 (±14.3) years, and at the follow-up survey, so was 51.4 (±14.7) years for women and 55.7 (±14.5) years for men, and the differences in mean age between women and men at baseline and follow-up were statistically significant ($p < 0.001$).

At the time of the follow-up survey, membership, or participation in WORTH Yetu was 10.1% of all the caregivers analysed. Since WORTH Yetu was a USAID Kizazi Kipya project-supported livelihoods programme, membership was at 0.0% at the baseline because there were no project services before enrollment. Further details regarding other background characteristics of the caregivers at the baseline and at the follow-up surveys in Table 1, and disaggregation of the same characteristics by gender is presented as supporting information in S1 Table.

## WORTH Yetu members' and non-members' characteristics at baseline

Table 2 shows the baseline characteristics of the OVC caregivers who were members and non-members of WORTH Yetu. While members and non-members of WORTH Yetu were similar in some baseline characteristics, the results revealed notably large differences in most characteristics, including place of residence, family size, and age. The observed differences in the baseline characteristics confirmed the existence of selection bias inherent in programmes that are non-experimental by design [3].

## Levels of household hunger at baseline and at follow-up

There was a significant change ($p < 0.001$) in levels of household hunger between baseline and follow-up surveys. As shown in Table 3, households with little to no hunger (food secure) increased from 25.7% (25.2% women and 27.0% men) at baseline to 31.3% (30.8% women and 32.6% men) at the follow-up survey; moderate hunger declined negligibly from 65.5% (66.1% women and 63.9% men) at baseline to 65.4% (65.8% women and 64.4% men) at the follow-up; and severe hunger declined from 8.8% (8.7% women and 9.1% men) at baseline to 3.3% (3.4% women and 3.0% men) at the endline survey. The observed positive changes in the levels of household hunger by gender, appeared to be more among men than women, especially at the follow-up survey.

## Levels of household hunger at follow-up by WORTH Yetu membership status

Overall, 31.3%, 65.4%, and 3.3% of the caregivers were in households with little to no hunger (food secure), moderate hunger, and severe hunger, respectively at the follow-up survey. These percentages were significantly different by WORTH Yetu membership status ($p < 0.001$), whereby, the percent of caregivers in little to no hunger households increased from 30.5% (30.0% women and 31.8% men) among non-members to 38.4% (37.9% women and 39.6% men) among WORTH Yetu members; moderate hunger declined from 66.1% (66.6% women and 65.2% men) among non-members to 58.5% (59.1% women and 57.1% men) among WORTH Yetu members; and severe hunger declined from 3.4% (3.5% women and 3.0% men) among non-members to 3.1% (2.9% women and 3.4% men) among WORTH Yetu members (Table 4).

**Table 1. Frequency distribution of respondents at baseline and follow-up.**

| Characteristic | Baseline | | Follow-up | |
|---|---|---|---|---|
| | Number (n) | Percent (%) | Number (n) | Percent (%) |
| **OVERALL** | 249,655 | 100.0 | 247,638 | 100.0 |
| **WORTH Yetu** | | | | |
| Non-member | 249,655 | 100.0 | 222,531 | 89.9 |
| Member | 0 | 0.0 | 25,107 | 10.1 |
| **Gender** | | | | |
| Women | 174,678 | 70.0 | 173,244 | 70.0 |
| Men | 74,977 | 30.0 | 74,394 | 30.0 |
| **Age** | | | | |
| 18–29 years | 15,226 | 6.1 | 9,223 | 3.7 |
| 30–39 years | 52,955 | 21.2 | 39,981 | 16.1 |
| 40–49 years | 74,118 | 29.7 | 70,993 | 28.7 |
| 50–59 years | 46,742 | 18.7 | 55,656 | 22.5 |
| 60+ years | 60,614 | 24.3 | 71,785 | 29.0 |
| **Marital status** | | | | |
| Married or living together | 126,135 | 50.5 | 134,366 | 54.3 |
| Divorced or separated | 37,971 | 15.2 | 37,571 | 15.2 |
| Widow or widower | 67,174 | 26.9 | 60,148 | 24.3 |
| Single or unmarried | 18,375 | 7.4 | 15,553 | 6.3 |
| **Education** | | | | |
| Never attended | 52,989 | 21.2 | 52,476 | 21.2 |
| Primary | 188,214 | 75.4 | 186,752 | 75.4 |
| Secondary+ | 8,452 | 3.4 | 8,410 | 3.4 |
| **HIV status** | | | | |
| Negative | 99,593 | 39.9 | 98,604 | 39.8 |
| Positive | 92,608 | 37.1 | 92,159 | 37.2 |
| Unknown | 57,454 | 23.0 | 56,875 | 23.0 |
| **Place of residence** | | | | |
| Rural | 140,773 | 56.4 | 139,246 | 56.2 |
| Urban | 108,882 | 43.6 | 108,392 | 43.8 |
| **Health insurance** | | | | |
| Uninsured | 219,924 | 88.1 | 208,083 | 84.0 |
| Insured | 29,731 | 11.9 | 39,555 | 16.0 |
| **Disability status** | | | | |
| Not disabled | 241,557 | 96.8 | 239,585 | 96.8 |
| Disabled | 8,098 | 3.2 | 8,053 | 3.3 |
| **Family size** | | | | |
| 2–3 people | 156,890 | 62.8 | 155,316 | 62.7 |
| 4–6 people | 81,960 | 32.8 | 81,576 | 32.9 |
| 7+ people | 10,805 | 4.3 | 10,746 | 4.3 |

## SES at baseline and at follow-up

There was a significant change in SES between baseline and follow-up surveys overall and for both women and men ($p < 0.001$). Briefly, caregivers in the lowest wealth quintile declined from 34.8% (39.0% women and 25.2% men) at baseline to 30.8% (34.2% women and 22.9% men) at the follow-up survey; the richest wealth quintile did not change and remained at 19.6%, but differences between women and men existed– 15.4% among women and 29.2%

**Table 2. Baseline characteristics of OVC caregivers who were members and non-members of WORTH Yetu at the follow-up.**

| Baseline characteristics | Member of WORTH Yetu | | Non-member of WORTH Yetu | | Missing WORTH Yetu status (LTFU) | |
|---|---|---|---|---|---|---|
| | n | % | n | % | n | % |
| **Overall** | **25,103** | **100.0** | **222,344** | **100.0** | **2,208** | **100.0** |
| **Sex** | | | | | | |
| Female | 17,981 | 71.6 | 155,113 | 69.8 | 1,584 | 71.7 |
| Male | 7,122 | 28.4 | 67,231 | 30.2 | 624 | 28.3 |
| **Age** | | | | | | |
| 18–29 years | 861 | 3.4 | 14,272 | 6.4 | 93 | 4.2 |
| 30–39 years | 4,076 | 16.2 | 48,491 | 21.8 | 388 | 17.6 |
| 40–49 years | 7,161 | 28.5 | 66,305 | 29.8 | 652 | 29.5 |
| 50–59 years | 5,317 | 21.2 | 40,971 | 18.4 | 454 | 20.6 |
| 60+ years | 7,688 | 30.6 | 52,305 | 23.5 | 621 | 28.1 |
| **Marital status** | | | | | | |
| Married or living together | 12,337 | 49.2 | 112,542 | 50.6 | 1,256 | 56.9 |
| Divorced or separated | 3,454 | 13.8 | 34,235 | 15.4 | 282 | 12.8 |
| Widow or widower | 8,064 | 32.1 | 58,550 | 26.3 | 560 | 25.4 |
| Single or unmarried | 1,248 | 5.0 | 17,017 | 7.7 | 110 | 5.0 |
| **Education** | | | | | | |
| Never attended | 6,334 | 25.2 | 46,125 | 20.7 | 530 | 24.0 |
| Primary | 18,161 | 72.4 | 168,435 | 75.8 | 1,618 | 73.3 |
| Secondary+ | 608 | 2.4 | 7,784 | 3.5 | 60 | 2.7 |
| **HIV status** | | | | | | |
| Negative | 9,832 | 39.2 | 88,634 | 39.9 | 1,127 | 51.0 |
| Positive | 8,096 | 32.3 | 84,040 | 37.8 | 472 | 21.4 |
| Unknown | 7,175 | 28.6 | 49,670 | 22.3 | 609 | 27.6 |
| **Place of residence** | | | | | | |
| Rural | 18,711 | 74.5 | 120,427 | 54.2 | 1,635 | 74.1 |
| Urban | 6,392 | 25.5 | 101,917 | 45.8 | 573 | 26.0 |
| **Health insurance** | | | | | | |
| Uninsured | 21,052 | 83.9 | 196,935 | 88.6 | 1,937 | 87.7 |
| Insured | 4,051 | 16.1 | 25,409 | 11.4 | 271 | 12.3 |
| **Disability status** | | | | | | |
| Not disabled | 24,192 | 96.4 | 215,208 | 96.8 | 2,157 | 97.7 |
| Disabled | 911 | 3.6 | 7,136 | 3.2 | 51 | 2.3 |
| **Family size** | | | | | | |
| 2–3 people | 13,150 | 52.4 | 141,996 | 63.9 | 1,744 | 79.0 |
| 4–6 people | 10,315 | 41.1 | 71,240 | 32.0 | 405 | 18.3 |
| 7+ people | 1,638 | 6.5 | 9,108 | 4.1 | 59 | 2.7 |

among men at baseline; and 15.9% among women and 28.4% among men at the follow-up survey. More details about changes in SES between baseline and follow-up are presented in Table 5.

## SES at follow-up by WORTH Yetu status

Table 6 compares SES at the follow-up survey between WORTH Yetu members and non-members, for both women and men. Findings show that the lowest wealth quintile declined from 32.4% among non-members to 16.5% among WORTH Yetu members and the richest wealth quintile increased from 18.9% (15.1% women and 27.6% men) among non-members to

**Table 3. Percent and corresponding 95% confidence interval (CI) of OVC caregivers in different levels of household hunger at baseline and at follow-up, disaggregated by gender.**

| | Baseline | | | Follow-up | | |
|---|---|---|---|---|---|---|
| | Men (n = 74,977) % (95% CI) | Women (n = 174,678) % (95% CI) | All (n = 249,655) % (95% CI) | Men (n = 74,394) % (95% CI) | Women (n = 173,244) % (95% CI) | All (n = 247,638) % (95% CI) |
| **Household hunger** | | | | | | |
| Little to no hunger | 27.0 (26.7, 27.3) | 25.2 (25.0, 25.4) | 25.7 (25.6, 25.9) | 32.6 (32.2, 32.9) | 30.8 (30.6, 31.0) | 31.3 (31.1, 31.5) |
| Moderate hunger | 63.9 (63.6, 64.3) | 66.1 (65.9, 66.3) | 65.5 (65.3, 65.6) | 64.4 (64.0, 64.7) | 65.8 (65.6, 66.0) | 65.4 (65.2, 65.5) |
| Severe hunger | 9.1 (8.9, 9.3) | 8.7 (8.6, 8.8) | 8.8 (8.7, 8.9) | 3.0 (2.91, 3.16) | 3.4 (3.36, 3.53) | 3.3 (3.25, 3.39) |

%: The percentages add up to 100 column-wise. The common denominator is the number of OVC caregivers (n) indicated in each column label, eg, n = 74,977 for men at the baseline. The numerator is the number of caregivers in each household hunger category. So, each of the presented percentages against each household hunger category across all the columns was computed as (numerator/denominator)*100.

CI: Clopper–Pearson's confidence interval.

26.2% (22.4% women and 35.7% men) among WORTH Yetu members at the follow-up survey. The other wealth quintiles (i.e., second, middle, and fourth) changed positively in favour of WORTH Yetu members with differences between women and men.

## Results of multivariate analysis

**Impact of WORTH Yetu on household hunger.** In the multivariate analysis (Table 7), after adjusting for other factors, namely, sex, education, marital status, age, health insurance, place of residence, disability status, and family size, the study found that:

There was a significant decline in the severity of household hunger by 33.3% among non-members of WORTH Yetu, but the decline became as large as 46.4% among WORTH Yetu members at the follow-up compared to the situation at the baseline (non-members at follow-up: aOR = 0.667, 95% CI [0.659, 0.676]; WORTH Yetu member at follow-up: aOR = 0.536, 95% CI [0.521, 0.553]) (Table 7, Model 1).

Regarding the caregivers' gender, men were significantly 10.7% more likely to experience more severe forms of household hunger compared to women (aOR = 1.107, 95% CI [1.089, 1.126]) (Table 7, Model 1).

**Table 4. Percent and corresponding 95% confidence interval (CI) of OVC caregivers in different levels of household hunger at follow-up by WORTH Yetu status, disaggregated by gender.**

| | WORTH Yetu Member | | | Non-Member | | | OVERALL | | |
|---|---|---|---|---|---|---|---|---|---|
| | Men (n = 7,123) % (95% CI) | Women (n = 17,984) % (95% CI) | All (n = 25,107) % (95% CI) | Men (n = 67,271) % (95% CI) | Women (n = 155,260) % (95% CI) | All (n = 222,531) % (95% CI) | Men (n = 74,394) % (95% CI) | Women (n = 173,244) % (95% CI) | All (n = 247,638) % (95% CI) |
| **Household hunger** | | | | | | | | | |
| Little to no hunger | 39.6 (38.4, 40.7) | 37.9 (37.2, 38.7) | 38.4 (37.8, 39.0) | 31.8 (31.5, 32.2) | 29.9 (29.7, 30.2) | 30.5 (30.3, 30.7) | 32.6 (32.2, 32.9) | 30.8 (30.6, 31.0) | 31.3 (31.1, 31.5) |
| Moderate hunger | 57.1 (55.9, 58.2) | 59.1 (58.4, 59.8) | 58.5 (57.9, 59.2) | 65.2 (64.8, 65.5) | 66.6 (66.3, 66.8) | 66.1 (65.9, 66.3) | 64.4 (64.0, 64.7) | 65.8 (65.6, 66.0) | 65.4 (65.2, 65.5) |
| Severe hunger | 3.4 (2.9, 3.8) | 2.9 (2.69, 3.19) | 3.1 (2.85, 3.28) | 3.0 (2.9, 3.1) | 3.5 (3.41, 3.59) | 3.4 (3.28, 3.43) | 3.0 (2.91, 3.16) | 3.4 (3.36, 3.53) | 3.3 (3.25, 3.39) |

%: The percentages add up to 100 column-wise. The common denominator is the number of OVC caregivers (n) indicated in each column label, eg, n = 7,123 for men who are members of WORTH Yetu. The numerator is the number of caregivers in each household hunger category. So, each of the presented percentages against each household hunger category across all the columns was computed as (numerator/denominator)*100.

CI: Clopper–Pearson's confidence interval.

**Table 5. Percent and corresponding 95% confidence interval of OVC caregivers in different categories (ie, wealth quintiles) of household socioeconomic status (SES) at baseline and at follow-up, disaggregated by gender.**

| | Baseline | | | Follow-up | | |
|---|---|---|---|---|---|---|
| | Men (n = 74,977) % (95% CI) | Women (n = 174,678) % (95% CI) | All (n = 249,655) % (95% CI) | Men (n = 74,394) % (95% CI) | Women (n = 173,244) % (95% CI) | All (n = 247,638) % (95% CI) |
| **SES** | | | | | | |
| Lowest (Q1) | 25.1 (24.8, 25.5) | 39.0 (38.8, 39.2) | 34.8 (34.7, 35.0) | 22.9 (22.6, 23.2) | 34.2 (34.0, 34.4) | 30.8 (30.6, 31.0) |
| Second (Q2) | 6.7 (6.6, 6.9) | 9.4 (9.3, 9.6) | 8.6 (8.5, 8.7) | 9.1 (8.9, 9.3) | 10.3 (10.2, 10.5) | 10.0 (9.8, 10.1) |
| Middle (Q3) | 17.9 (17.7, 18.2) | 18.1 (17.9, 18.3) | 18.1 (17.9, 18.2) | 27.3 (27.0, 27.7) | 30.5 (30.3, 30.7) | 29.6 (29.4, 29.7) |
| Fourth (Q4) | 21.0 (20.7, 21.3) | 18.1 (17.9, 18.2) | 18.9 (18.8, 19.1) | 12.3 (12.1, 12.5) | 9.0 (8.9, 9.2) | 10.0 (9.9, 10.1) |
| Richest (Q5) | 29.2 (28.9, 29.5) | 15.4 (15.3, 15.6) | 19.6 (19.4, 19.7) | 28.4 (28.0, 28.7) | 15.9 (15.7, 16.1) | 19.6 (19.5, 19.8) |

%: The percentages add up to 100 column-wise. The common denominator is the number of caregivers (n) indicated in each column, eg, n = 74,977 for men at the baseline. The numerator is the number of caregivers in each SES category (not indicated). So, each of the presented percentages against each SES category across all the columns was computed as (numerator/denominator)*100.

CI: Clopper–Pearson's confidence interval.

In the disaggregated models, the likelihood of experiencing more severe forms of household hunger was 32.5% less likely among women who were not yet members in WORTH Yetu at the follow-up survey (aOR = 0.674, 95% CI [0.664, 0.685]), but again this increased to 45.7% among women who were WORTH Yetu members at the follow-up (aOR = 0.543, 95% CI [0.524, 0.563]) than the situation at the baseline (Table 7, Model 2).

For men, those who were non-members at the follow-up survey were 35% less likely to experience more severe forms of household hunger (aOR = 0.650, 95% CI [0.635, 0.665]), but the likelihood increased to 47.5% among men who were WORTH Yetu members at the follow-up survey compared to the situation at baseline (aOR = 0.525, 95% CI [0.497, 0.556]) (Table 7, Model 3).

**Table 6. Percent and corresponding 95% confidence interval (CI) of OVC caregivers in different categories (ie, wealth quintiles) of household socioeconomic status (SES) at the follow-up survey by WORTH Yetu membership status, disaggregated by gender.**

| | WORTH Yetu Member | | | Non-Member | | | OVERALL | | |
|---|---|---|---|---|---|---|---|---|---|
| | Men (n = 7,123) % (95% CI) | Women (n = 17,984) % (95% CI) | All (n = 25,107) % (95% CI) | Men (n = 67,271) % (95% CI) | Women (n = 155,260) % (95% CI) | All (n = 222,531) % (95% CI) | Men (n = 74,394) % (95% CI) | Women (n = 173,244) % (95% CI) | All (n = 247,638) % (95% CI) |
| **SES** | | | | | | | | | |
| Lowest | 12.6 (11.8, 13.4) | 18.1 (17.5, 18.6) | 16.5 (16.1, 17.0) | 24.0 (23.7, 24.3) | 36.1 (35.9, 36.3) | 32.4 (32.2, 32.6) | 22.9 (22.6, 23.2) | 34.2 (34.0, 34.4) | 30.8 (30.6, 31.0) |
| Second | 8.2 (7.6, 8.8) | 11.3 (10.9, 11.8) | 10.4 (10.1, 10.8) | 9.2 (9.0, 9.4) | 10.2 (10.1, 10.4) | 9.9 (9.8, 10.0) | 9.1 (8.9, 9.3) | 10.3 (10.2, 10.5) | 10.0 (9.8, 10.1) |
| Middle | 27.3 (26.2, 28.3) | 33.6 (32.9, 34.3) | 31.8 (31.2, 32.4) | 27.3 (27.0, 27.7) | 30.2 (29.9, 30.4) | 29.3 (29.1, 29.5) | 27.3 (27.0, 27.7) | 30.5 (30.3, 30.7) | 29.6 (29.4, 29.7) |
| Fourth | 16.3 (15.4, 17.1) | 14.6 (14.1, 15.1) | 15.1 (14.6, 15.5) | 11.9 (11.6, 12.1) | 8.4 (8.2, 8.5) | 9.4 (9.3, 9.6) | 12.3 (12.1, 12.5) | 9.0 (8.9, 9.2) | 10.0 (9.9, 10.1) |
| Richest | 35.7 (34.6, 36.8) | 22.4 (21.8, 23.0) | 26.2 (25.6, 26.7) | 27.6 (27.3, 27.9) | 15.1 (15.0, 15.3) | 18.9 (18.7, 19.1) | 28.4 (28.0, 28.7) | 15.9 (15.7, 16.1) | 19.6 (19.5, 19.8) |

%: The percentages add up to 100 column-wise. The common denominator is the number of OVC caregivers (n) indicated in each column label, eg, n = 7,123 for men who are members of WORTH Yetu. The numerator is the number of caregivers in each SES category (not indicated). So, each of the presented percentages for each SES category across all the columns was computed as (numerator/denominator)*100.

CI: Clopper–Pearson's confidence interval.

**Table 7. Multivariate mixed-effects ordinal logistic regression models with a DiD estimator of the impact of WORTH Yetu on household hunger among OVC caregivers in Tanzania.**

| Covariate | Model 1 (All Caregivers) (n = 497,293) | | Model 2 (Women) (n = 347,922) | | Model 3 (Men) (n = 149,371) | |
|---|---|---|---|---|---|---|
| | aOR | (95% CI) | aOR | 95% CI | aOR | 95% CI |
| **WORTH Yetu*Time** | | | | | | |
| Follow-up (vs. baseline) | 0.667*** | (0.659, 0.676) | 0.674*** | (0.664, 0.685) | 0.650*** | (0.635, 0.665) |
| Non-member at follow-up | 0.667*** | (0.659, 0.676) | 0.674*** | (0.664, 0.685) | 0.650*** | (0.635, 0.665) |
| WORTH Yetu member at follow-up | 0.536*** | (0.521, 0.553) | 0.543*** | (0.524, 0.563) | 0.525*** | (0.497, 0.556) |
| **Gender** | | | | | | |
| Women | 1.000 | — | — | — | — | — |
| Men | 1.107*** | (1.089, 1.126) | — | — | — | — |
| **Age** | | | | | | |
| 18–29 years | 1.000 | — | 1.000 | — | 1.000 | — |
| 30–39 years | 0.861*** | (0.831, 0.892) | 0.853** | (0.821, 0.887) | 0.911* | (0.831, 1.000) |
| 40–49 years | 0.805*** | (0.777, 0.833) | 0.794*** | (0.765, 0.825) | 0.878** | (0.802, 0.961) |
| 50–59 years | 0.753*** | (0.726, 0.780) | 0.729*** | (0.701, 0.759) | 0.866** | (0.785, 0.942) |
| 60+ years | 0.713*** | (0.687, 0.739) | 0.694*** | (0.666, 0.722) | 0.805*** | (0.735, 0.881) |
| **Marital status** | | | | | | |
| Married or living together | 1.000 | — | 1.000 | — | 1.000 | — |
| Divorced or separated | 1.265*** | (1.240, 1.290) | 1.282*** | (1.253, 1.311) | 1.226*** | (1.179, 1.275) |
| widow or widower | 1.158*** | (1.138, 1.178) | 1.167*** | (1.144, 1.191) | 1.163*** | (1.120, 1.207) |
| Single or unmarried | 1.249*** | (1.214, 1.285) | 1.258*** | (1.219, 1.298) | 1.210*** | (1.130, 1.295) |
| **Education** | | | | | | |
| Never attended | 1.000 | — | 1.000 | — | 1.000 | — |
| Primary | 0.771*** | (0.757, 0.785) | 0.752*** | (0.736, 0.769) | 0.808*** | (0.785, 0.836) |
| Secondary+ | 0.628*** | (0.602, 0.655) | 0.609*** | (0.578, 0.640) | 0.673*** | (0.623, 0.726) |
| **HIV status** | | | | | | |
| Negative | 1.000 | — | 1.000 | — | 1.000 | — |
| Positive | 0.748*** | (0.735, 0.760) | 0.800*** | (0.785, 0.816) | 0.639*** | (0.619, 0.658) |
| Unknown | 0.946*** | (0.928, 0.964) | 0.962** | (0.940, 0.984) | 0.918*** | (0.889, 0.947) |
| **Place of residence** | | | | | | |
| Rural | 1.000 | — | 1.000 | — | 1.000 | — |
| Urban | 2.043*** | (2.011, 2.074) | 2.161*** | (2.122, 2.200) | 1.787*** | (1.736, 1.839) |
| **Health insurance** | | | | | | |
| Uninsured | 1.000 | — | 1.000 | — | 1.000 | — |
| Insured | 0.594*** | (0.583, 0.606) | 0.606*** | (0.592, 0.620) | 0.573*** | (0.554, 0.593) |
| **Disability status** | | | | | | |
| Not disabled | 1.000 | — | 1.000 | — | 1.000 | — |
| Disabled | 1.301*** | (1.249, 1.355) | 1.331*** | (1.263, 1.402) | 1.246*** | (1.168, 1.330) |
| **Family size** | | | | | | |
| 2–3 people | 1.000 | — | 1.000 | — | 1.000 | — |
| 4–6 people | 1.107*** | (1.090, 1.124) | 1.133*** | (1.112, 1.154) | 1.061*** | (1.031, 1.092) |
| 7+ people | 1.259*** | (1.216, 1.305) | 1.339*** | (1.280, 1.400) | 1.146*** | (1.081, 1.214) |
| **/Cut1** | -1.491 | (-1.530, -1.451) | -1.456 | (-1.500, -1.412) | -1.572 | (-1.667, -1.477) |
| **/Cut2** | 2.819 | (2.778, 2.860) | 2.897 | (2.850, 2.943) | 2.638 | (2.541, 2.735) |
| **Var(constant)** | 0.821 | (0.794, 0.848) | 0.822 | (0.790, 0.855) | 0.807 | (0.760, 0.857) |

(*Continued*)

**Table 7.** (Continued)

| Covariate | Model 1 (All Caregivers) (n = 497,293) | | Model 2 (Women) (n = 347,922) | | Model 3 (Men) (n = 149,371) | |
|---|---|---|---|---|---|---|
| | aOR | (95% CI) | aOR | 95% CI | aOR | 95% CI |
| **ICC** | 0.200 | (0.194, 0.205) | 0.200 | (0.194, 0.206) | 0.197 | (0.188, 0.207) |

aOR: Adjusted Odds Ratio

CI: Confidence Interval

DiD: Difference in Differences

ICC: Intraclass correlation coefficient

OVC: Orphans and vulnerable children.

Significance

***p<0.001

**p<0.050

*p<0.100

Number of caregivers = 249,846; Number of observations per caregiver: min = 1, average = 2.0, max = 2, total number of observations analysed = 497,293

Log likelihood = -385,102.71

Number of women caregivers = 174,828; number of observations per woman: min = 1, average = 2.0, max = 2, total number of observations analysed = 347,922.

Log likelihood = -267,342.29

Number of male caregivers = 75,018; number of observations per caregiver: min = 1, average = 2.0, max = 2, total number of observations analysed = 149,371

Log likelihood = -117,550.24

Most of the other factors which influenced the level of household hunger were similar by gender, except age, whereby the likelihood to experience the more severe forms of household hunger declined in a dose-response fashion in all age groups above 29 years for women, but for men, the decline was significant only in age groups above 39 years. This suggested that the protective effect of age against household hunger was not equally felt between women and men, implying that women were more likely to be food secure at a younger age than men (Table 7, Models 2 and 3).

The intraclass correlation coefficient (ICC) for each of Models 1, 2, and 3 in Table 7 was 20%, representing the amount of correlation between observations of the same caregiver. For each of the models, the *p*-value from the likelihood-ratio (LR) test that a variance component was zero was <0.001, emphasizing that fitting the regression models while recognizing the clustering of observations within caregivers was statistically more appropriate than fitting the standard models.

## Impact of WORTH Yetu on SES

Table 8 presents the impact of WORTH Yetu on SES, after addressing selection bias in terms of gender, age, marital status, education, HIV status, place of residence, health insurance, disability status, and family size.

Results reveal (in Table 8, Model 1) that non-members in WORTH Yetu were 14.9% less likely to be in higher wealth quintiles at the follow-up (aOR = 0.851, 95% CI [0.842, 0.861]), while WORTH Yetu members were 15.9% more likely to be in higher wealth quintiles at the follow-up compared to the situation at the baseline (aOR = 1.159, 95% CI [1.128, 1.190]).

By gender, men were 53.6% more likely to be in higher wealth quintiles than their women counterparts (aOR = 1.536, 95% CI [1.511, 1.561]) (Table 8, Model 1).

In the disaggregated analysis, women who were not in WORTH Yetu were 12.6% less likely to be in higher wealth quintiles at the follow-up (aOR = 0.874, 95% CI [0.862, 0.886]), while women who were WORTH Yetu members were 20.8% more likely to be in higher wealth

**Table 8. Multivariate mixed-effects ordinal logistic regression models with a DiD estimator of the impact of WORTH Yetu on household socioeconomic status (SES) among OVC caregivers in Tanzania.**

| Covariate | Model 1 (All Caregivers) (n = 497,293) | | Model 2 (Women) (n = 347,922) | | Model 3 (Men) (n = 149,371) | |
|---|---|---|---|---|---|---|
| | aOR | (95% CI) | aOR | 95% CI | aOR | 95% CI |
| **WORTH Yetu\*Time** | | | | | | |
| Follow-up (vs. baseline) | 0.851*** | (0.842, 0.861) | 0.874*** | (0.862, 0.886) | 0.811*** | (0.795, 0.828) |
| Non-member at follow-up | 0.851*** | (0.842, 0.861) | 0.874*** | (0.862, 0.886) | 0.811*** | (0.795, 0.828) |
| WORTH Yetu member at follow-up | 1.159*** | (1.128, 1.190) | 1.208*** | (1.170, 1.247) | 1.046* | (0.995, 1.101) |
| **Gender** | | | | | | |
| Women | 1.000 | — | — | — | — | — |
| Men | 1.536*** | (1.511, 1.561) | — | — | — | — |
| **Age** | | | | | | |
| 18–29 years | 1.000 | — | 1.000 | — | 1.000 | — |
| 30–39 years | 1.234*** | (1.191, 1.277) | 1.236*** | (1.190, 1.284) | 1.150** | (1.055, 1.254) |
| 40–49 years | 1.613*** | (1.559, 1.670) | 1.624*** | (1.563, 1.686) | 1.439*** | (1.323, 1.567) |
| 50–59 years | 2.076*** | (2.004, 2.151) | 2.156*** | (2.072, 2.243) | 1.721*** | (1.580, 1.875) |
| 60+ years | 2.339*** | (2.257, 2.424) | 2.391*** | (2.296, 2.489) | 1.991*** | (1.829, 2.168) |
| **Marital status** | | | | | | |
| Married or living together | 1.000 | — | 1.000 | — | 1.000 | — |
| Divorced or separated | 0.617*** | (0.606, 0.629) | 0.620*** | (0.607, 0.633) | 0.618*** | (0.596, 0.640) |
| widow or widower | 0.671*** | (0.660, 0.681) | 0.689*** | (0.676, 0.702) | 0.591*** | (0.571, 0.611) |
| Single or unmarried | 0.453*** | (0.441, 0.466) | 0.456*** | (0.442, 0.470) | 0.457*** | (0.429, 0.486) |
| **Education** | | | | | | |
| Never attended | 1.000 | — | 1.000 | — | 1.000 | — |
| Primary | 1.152*** | (1.132, 1.172) | 1.150*** | (1.126, 1.174) | 1.184*** | (1.145, 1.223) |
| Secondary+ | 1.192*** | (1.143, 1.244) | 1.170*** | (1.112, 1.231) | 1.263*** | (1.172, 1.361) |
| **HIV status** | | | | | | |
| Negative | 1.000 | — | 1.000 | — | 1.000 | — |
| Positive | 0.961*** | (0.945, 0.977) | 0.914*** | (0.896, 0.932) | 1.084*** | (1.052, 1.117) |
| Unknown | 0.988 | (0.970, 1.006) | 0.999 | (0.977, 1.022) | 0.965** | (0.935, 0.996) |
| **Place of residence** | | | | | | |
| Rural | 1.000 | — | 1.000 | — | 1.000 | — |
| Urban | 0.202*** | (0.199, 0.205) | 0.187*** | (0.183, 0.190) | 0.242*** | (0.236, 0.250) |
| **Health insurance** | | | | | | |
| Uninsured | 1.000 | — | 1.000 | — | 1.000 | — |
| Insured | 1.879*** | (1.847, 1.913) | 1.863*** | (1.824, 1.902) | 1.904*** | (1.844, 1.965) |
| **Disability status** | | | | | | |
| Not disabled | 1.000 | — | 1.000 | — | 1.000 | — |
| Disabled | 0.796*** | (0.766, 0.828) | 0.831*** | (0.790, 0.874) | 0.759*** | (0.713, 0.807) |
| **Family size** | | | | | | |
| 2–3 people | 1.000 | — | 1.000 | — | 1.000 | — |
| 4–6 people | 1.106*** | (1.090, 1.123) | 1.092*** | (1.072, 1.112) | 1.126*** | (1.096, 1.158) |
| 7+ people | 1.557*** | (1.504, 1.612) | 1.498*** | (1.435, 1.565) | 1.642*** | (1.550, 1.739) |
| **/Cut1** | -1.100 | (-1.138, -1.062) | -1.160 | (-1.203, -1.117) | -1.466 | (-1.555, -1.376) |
| **/Cut2** | -0.533 | (-0.571, -0.495) | -0.577 | (-0.619, -0.534 | -0.943 | (-1.033, -0.854) |
| **/Cut3** | 0.859 | (0.821, 0.897 | 0.869 | (0.826, 0.911) | 0.333 | (0.244, 0.422) |
| **/Cut4** | 1.887 | (1.848, 1.926) | 1.931 | (1.887, 1.975) | 1.305 | (1.216, 1.394) |
| **Var(constant)** | 0.212 | (1.186, 1.239) | 1.226 | (1.194, 1.259) | 1.169 | (1.123, 1.217) |

*(Continued)*

**Table 8.** (Continued)

| Covariate | Model 1 (All Caregivers) (n = 497,293) | | Model 2 (Women) (n = 347,922) | | Model 3 (Men) (n = 149,371) | |
|---|---|---|---|---|---|---|
| | aOR | (95% CI) | aOR | 95% CI | aOR | 95% CI |
| ICC | 0.269 | (0.265, 0.274) | 0.272 | (0.266, 0.277) | 0.262 | (0.254, 0.270) |

aOR: Adjusted Odds Ratio

CI: Confidence Interval

DiD: Difference in Differences

ICC: Intraclass correlation coefficient

OVC: Orphans and vulnerable children.

Significance

***p<0.001

**p<0.050

*p<0.100

Number of caregivers = 249,846; Number of observations per caregiver: min = 1, average = 2.0, max = 2, total number of observations analysed = 497,293

Log likelihood = -700,447.5

Number of women caregivers = 174,828; number of observations per woman: min = 1, average = 2.0, max = 2, total number of observations analysed = 347,922.

Log likelihood = -483,212.13

Number of male caregivers = 75,018; number of observations per caregiver: min = 1, average = 2.0, max = 2, total number of observations analysed = 149,371

Log likelihood = -216,605.73

quintiles at the follow-up (aOR = 1.208, 95% CI [1.170, 1.247]) compared to the situation at the baseline (Table 8, Model 2). For men (Table 8, Model 3), non-members of WORTH Yetu were 18.9% less likely to be in higher wealth quintiles (aOR = 0.811, 95% CI [0.795, 0.828]), while men who were WORTH Yetu members were 4.6% more likely to be in higher wealth quintiles at the follow-up (aOR = 1.046, 95% CI [0.995, 1.101]) compared to the situation at the baseline. This effect was not statistically significant at the 5% level but indicates that the WORTH Yetu intervention was protective against the loss of household assets.

The ICC for the three models in Table 8 was 27% for each of Model 1 and Model 2, and 26% for Model 3. Again, this indicated the degree of correlation of observations of the same caregiver, favouring the use of multilevel models which account for within-cluster correlations over standard models [49]. The LR test indicated that the variance component in each of the models was not zero ($p < 0.001$), hence the multilevel models were appropriately used in this case over standard models.

## Discussion

This study investigated the significance of gender disaggregation in impact evaluation of non-experimental livelihood interventions, based on the analysis of WORTH Yetu impact on household hunger, and SES among OVC caregivers in Tanzania. For each of the two outcomes, a multivariate model was fitted for all the caregivers, after which two separate models, one for women and another for men, followed. In each of the models, potential confounders controlled for to account for selection bias were age, marital status, education attained, health insurance, HIV status, place of residence, disability status, and family size. After adjusting for these factors, the overall findings revealed a significant decline in household hunger by 46.4% among WORTH Yetu members at the follow-up compared to the situation at the baseline (aOR = 0.536, 95% CI [0.521, 0.553], $p < 0.001$). In the gender disaggregated models (ie, within gender comparisons), the decline was 45.7% among women who were members of WORTH Yetu (aOR = 0.543, 95% CI [0.524, 0.563], $p < 0.001$) and 47.5% among men who were

members of WORTH Yetu (aOR = 0.525, 95% CI [0.497, 0.556], $p < 0.001$) at the follow-up compared to their respective situations at the baseline. These findings are consistent with those from experimental programmes such as the *Chuma na Uchizi*, a livelihood intervention that reduced food insecurity among PLHIV in Zambia [50].

Regarding household SES, the odds of being in higher wealth quintiles was significantly 1.159 times higher among WORTH Yetu members at the follow-up compared to the situation at the baseline (aOR = 1.159, 95% CI [1.128, 1.190], $p < 0.001$). After disaggregating the analysis by gender (ie, within gender comparisons), the odds of being in higher wealth quintiles was significantly 1.208 times higher among women who were WORTH Yetu members (aOR = 1.208, 95% CI [1.170, 1.247], $p < 0.001$) and 1.046 times higher among men who were WORTH Yetu members (aOR = 1.046, 95% CI [0.995, 1.101], $p = 0.080$) at the follow-up compared to their respective situations at the baseline. The intervention's impact among men was positive, but not statistically significant. Of note is that, without WORTH Yetu (ie, intervention non-recipients), the likelihood of being in higher wealth quintiles was significantly declining overtime in the overall model ($p < 0.001$), as well as in the women's ($p < 0.001$) and men's ($p < 0.001$) models. These findings suggested that while the WORTH Yetu intervention facilitated household asset acquisition among members overall, and more so among women, the intervention protected household asset loss (with no significant evidence of improved asset acquisition) overtime among men.

Between-gender comparison showed that, men and women were significantly different with respect to WORTH Yetu impact on both outcomes–household hunger, and SES. Specifically, men were significantly more likely than women to be in more severe forms of household hunger than women at the follow-up compared to the baseline situation. However, men were more likely than women to be in higher wealth quintiles than women at the follow-up survey than the situation at the baseline.

In addition, other factors, apart from the WORTH Yetu intervention, which influenced both household hunger and SES outcomes were not perfectly the same for men and women. Of course, many factors exerted a similar influence on women and men for both outcomes, but some were stronger for one gender than the other. For example, the likelihood of being in more severe forms of household hunger declined as age increased for all age groups above 29 years for women, but so was not the case until after age 39 years for men. In other words, age was a protective factor against household hunger, but the protection was stronger and felt early at younger ages in women than in men. Although the overall relationship of age and household hunger observed in the present study is consistent with other studies [51–54], a positive association between female gender and food security has been noted in some studies eg, [55]. Therefore, interventions aiming at addressing household hunger in vulnerable populations such as OVC caregivers should be designed in a gender-responsive manner, recognising that men may require additional support and strategies to optimize programme impacts.

Also, the influence of HIV status on SES was very different between women and men. Overall, caregivers LHIV were significantly less likely to be in higher wealth quintiles at the follow-up than those who were HIV negative. Those of unknown HIV status were similar to those who were HIV negative. While this observation was similar as that among women, for men, those who were HIV positive were significantly more likely to be in higher wealth quintiles than their HIV negative counterparts; and those of unknown HIV status were significantly less likely to be in higher wealth quintiles than those who were HIV negative. Although the underlying mechanism of these results is not clear, strategies to improve the outcomes among the intervention recipients should be tailored to their HIV status, so that those at a low chance of benefiting from the intervention are given more support as needed. Other factors, namely,

marital status, education, place of residence, disability status, and health insurance were discussed in the earlier study largely based on the same population [40].

All these differences between women and men emphasise that the genders are different, and indeed highlight the significance of gender disaggregation in impact evaluation of non-experimental livelihood or economic empowerment interventions. These disparities may, in part, stem from differences in access to resources including education, employment opportunities, and control over income. Additionally, traditional gender roles, which allocate varied responsibilities such as household chores for women and breadwinning for men within households and communities [56], contribute to shaping how individuals experience and respond to interventions. This is in line with the Realist Evaluation theory which posits that no intervention works everywhere and for everyone, which is why the focus should be to find what works, for whom, and why it does or does not work [35].

## Strengths and limitations

This study is based on a large sample size along a national-wide geographical coverage, permitting the results to be nationally representative. Also, statistical methods employed in the evaluation of the impact of WORTH Yetu on household hunger and SES are scientifically rigorous and addressed selection bias based on a wide range of potential confounders adjusted for in the multivariate models. This guarantees that the estimated impacts are as close to reality as possible, leaving a minimal possibility that the findings are due to chance or confounding.

Although many factors were adjusted for to address selection bias in this study, we acknowledge a possibility of residual confounding, especially due to factors which were not available for inclusion in the analysis.

## Conclusion

The present study found that WORTH Yetu reduced household hunger on one hand, and improved household SES on the other, with significant variations in the observed impacts between women and men. WORTH Yetu reduced the likelihood of being in more severe forms of household hunger by 46.4% overall, and 45.7% for women and 47.5% for men within the average follow-up period of 1.6 years from the baseline to the follow-up survey. With respect to SES, WORTH Yetu improved the likelihood of being in higher wealth quintiles by 15.9% overall, and by 20.8% for women and only 4.6% for men within the same period.

Between gender comparisons emphasised that while men were significantly more likely to experience severe forms of household hunger than women, men were more likely to be in higher wealth quintiles than women.

For SES, findings clearly suggest that the WORTH Yetu intervention was significantly effective in improving household SES, particularly for women. However, for men, the WORTH Yetu impact on SES was positive, but not statistically significant. A common observation in all the three models is that non-members of WORTH Yetu were significantly less likely to improve their SES at the follow-up compared to the situation at the baseline. This partly suggested that even if WORTH Yetu did not significantly improve SES for men, it was at least protective against further loss of household assets over time.

Without gender disaggregation, the observed differences between women and men with respect to the WORTH Yetu impact on household hunger and SES would not have been detected. Many similar studies have simply included gender or sex as one of the explanatory variables, eg, [34], for which in our case, we would have ended up with the odds of higher wealth quintiles being 1.536 times higher for men compared to women. But the deeper analysis uncovered further that women and men were significantly different with respect to the

WORTH Yetu impact on SES, whereby the gain in SES due to the WORTH Yetu intervention was significantly higher among women (when comparing women to women) than men (when comparing men to men).

Overall, women and men experienced the livelihood outcomes attributable to the WORTH Yetu intervention differently, highlighting the distinct nature of these populations in the context of economic empowerment programmes. These findings emphasize the importance of prioritising gender as a critical dimension in the design, delivery, and evaluation of livelihood programmes. Moreover, accelerating the coverage of the WORTH Yetu intervention is essential as a viable strategy to combat household hunger and enhance the socioeconomic wellbeing of families caring for OVC and other vulnerable populations. This may require strategies that are responsive to gender-specific needs and differences to maximise the gains of the interventions, eg, providing targeted support to female caregivers LHIV and male caregivers of undisclosed HIV status to enhance their SES etc. These results can likely be applied in similar contexts and settings to appropriately gauge impacts of similar programmes.

## Supporting information

**S1 Table. Frequency distribution of respondents at baseline and at follow-up, disaggregated by gender.**
(PDF)

## Acknowledgments

We are thankful to all efforts invested in this work to make it a success. Mr. Nemes A. Mallya (Senior Program Officer–Economic Strengthening) of Pact Tanzania is greatly acknowledged for constantly guiding us on ES matters.

## Author Contributions

**Conceptualization:** Amon Exavery.

**Data curation:** Amon Exavery, John Charles.

**Formal analysis:** Amon Exavery, Peter J. Kirigiti, Ramkumar T. Balan, John Charles.

**Funding acquisition:** Amon Exavery.

**Investigation:** Amon Exavery, Peter J. Kirigiti, Ramkumar T. Balan, John Charles.

**Methodology:** Amon Exavery, Peter J. Kirigiti, Ramkumar T. Balan, John Charles.

**Project administration:** Amon Exavery, Peter J. Kirigiti, Ramkumar T. Balan, John Charles.

**Resources:** Amon Exavery.

**Software:** Amon Exavery, John Charles.

**Supervision:** Peter J. Kirigiti, Ramkumar T. Balan, John Charles.

**Validation:** Amon Exavery, John Charles.

**Visualization:** Amon Exavery, Peter J. Kirigiti, Ramkumar T. Balan, John Charles.

**Writing – original draft:** Amon Exavery.

**Writing – review & editing:** Amon Exavery, Peter J. Kirigiti, Ramkumar T. Balan, John Charles.

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
