## [Decision Letter · Decision Letter 0]

15 Nov 2023

PONE-D-23-25792Multivariate mixed-effects ordinal logistic regression models with difference-in-differences estimator of the impact of WORTH Yetu on household hunger and socioeconomic status among OVC caregivers in TanzaniaPLOS ONE

Dear Dr. Exavery,

Thank you for submitting your manuscript to PLOS ONE. After careful consideration, we feel that it has merit but does not fully meet PLOS ONE’s publication criteria as it currently stands. Therefore, we invite you to submit a revised version of the manuscript that addresses the points raised during the review process.

ACADEMIC EDITOR:Your manuscript has been reviewed and the manuscript requires minor corrections before it can be submitted. Please addressing the following minor comments which have also been raised by the reviewers;In line 34 …you mention the use of  the follow-up data (2019-2020), the reader may be interested to know whether this was an endline data? Provide some bit of explanationBriefly explain why the mixed effects ordinal logistic regression models were the ones suited for the data.

We look forward to receiving your revised manuscript.

Kind regards,

Mpho Keetile, PhD

Academic Editor

PLOS ONE

3. Please include a copy of Table 6 which you refer to in your text on page 22.

Additional Editor Comments:

There are some minor comments raised by the reviewers (see them below) and once addressed the manuscript will be ready for publication. Address the comments and return the manuscript for final review

Reviewers' comments:

Reviewer's Responses to Questions

**Comments to the Author**

1. Is the manuscript technically sound, and do the data support the conclusions?

Reviewer #1: Yes

Reviewer #2: Yes

2. Has the statistical analysis been performed appropriately and rigorously? 

Reviewer #1: Yes

Reviewer #2: I Don't Know

3. Have the authors made all data underlying the findings in their manuscript fully available?

Reviewer #1: Yes

Reviewer #2: Yes

4. Is the manuscript presented in an intelligible fashion and written in standard English?

Reviewer #1: Yes

Reviewer #2: Yes

5. Review Comments to the Author

Reviewer #1: This manuscript presents an important and timely research topic that is of its kind in in many sub-regions on the African continent. Health services uptake in connection to economic needs is an important area that needs national exploration to capture relevance-embedded interventions aiming to empower community to improve livelihoods.

I have read the manuscript several times and have concluded that the authors conducted the study in a novel and appropriate manner adhering to research ethics guidelines and step-wise procedure. The methodology including the study design, settings, sampling, and data collection were thoroughly described in clear language. Only the following to take into account.

Methodology

On line 34 … the follow-up data (2019-2020) …. Was it the endline data? Can you clarify to make the reader understand better?

On line 153 it mentioned 25 Tanzania region, does the study included all regions. If not what was the exclusion criteria?

The statistical analysis in my opinion was robust and clearly presents the findings of the study that I believe many readers will understand and appreciate but I also believe that the write up of the results could be presented in a more step-by-step manners for most readers. The tables are presented very clearly nonetheless.

The discussion is well presented and reflects what the relationships of the findings are to existing literature.

Overall this is great piece of work that pave a way of evaluating embedded economic strengthening intervention to a parent intervention of improving access to social health care.

Reviewer #2: General: This is a very interesting study that recognizes the importance of gender equity in relation to social-economic outcomes of specific interventions. The study is timely since in Tanzania there is an increase in social economic programs targeting household members but there is no concrete plan to ensure that both men and women benefit equitably from these programs.

I have minor comments

Abstract

1. It is important that the abstract clearly indicates the geographic regions where the data was collected

2. The objectives of the study need to be clearly stipulated in the abstract and toward the end of the background section of the main paper

3. It is not clear in the abstract on who the WORTH YETU group was targeting, is it women or men?

4. When the author draws a conclusion that “the design, delivery, and evaluation of such programmes should happen in a gender responsive manner, recognizing that women and men are not the same with respect to the programmes”. This statement is too general, not very clear what it means that men and women are not the same with respect to the program’. The author may consider including a more specific implication of these interesting findings.

Main paper

Background

1. It would be important if the author could introduce the first paragraph by providing information on the current gender inequality index in Tanzania and some evidence on gender inequality practices as this would provide a contextual foundation of the topic. The methodological aspect could be introduced in the middle of the background section.

2. Also it is important to include the main purpose of the paper towards the end of the background section

Results

The author indicate that severity of household hunger declined by 45.7% among women (aOR = 0.543 [0.524, 0.563]) and 47.5% among men (aOR = 0.525 [0.497, 0.556]). I find this to be an interesting finding. It is important to highlight any possible determinants of these differences, including gender role dynamics and contextual explanations.

Conclusion

It is important for authors to indicate their advice clearly in the paper when proposing that . Overall, these findings imply that women and men are different with respect to 640 livelihood programmes, hence a need to prioritize gender as a critical dimension in the 641 design, delivery, and evaluation of the programmes”

6. PLOS authors have the option to publish the peer review history of their article (what does this mean?). If published, this will include your full peer review and any attached files.

Reviewer #1: No

Reviewer #2: No

---

## [Author Response · Author response to Decision Letter 0]

23 Nov 2023

Responses to all reviewers' and editor's comments are uploaded in one file named "Response to Reviewers"

---

## [Decision Letter · Decision Letter 1]

20 Mar 2024

Multivariate mixed-effects ordinal logistic regression models with difference-in-differences estimator of the impact of WORTH Yetu on household hunger and socioeconomic status among OVC caregivers in Tanzania

PONE-D-23-25792R1

Dear Dr. Exavery Amon,

We’re pleased to inform you that your manuscript has been judged scientifically suitable for publication and will be formally accepted for publication once it meets all outstanding technical requirements.

An invoice for payment will follow shortly after the formal acceptance. To ensure an efficient process, please log into Editorial Manager at Editorial Manager® , click the 'Update My Information' link at the top of the page, and double check that your user information is up-to-date. If you have any billing related questions, please contact our Author Billing department directly at authorbilling@plos.org.

Kind regards,

Dr Moses Katbi MD, MPH, MBA, DrPH

Academic Editor

PLOS ONE

Additional Editor Comments (optional):

Reviewers' comments:

Reviewer's Responses to Questions

**Comments to the Author**

1. If the authors have adequately addressed your comments raised in a previous round of review and you feel that this manuscript is now acceptable for publication, you may indicate that here to bypass the “Comments to the Author” section, enter your conflict of interest statement in the “Confidential to Editor” section, and submit your "Accept" recommendation.

Reviewer #1: All comments have been addressed

Reviewer #3: (No Response)

2. Is the manuscript technically sound, and do the data support the conclusions?

Reviewer #1: Yes

Reviewer #3: Yes

3. Has the statistical analysis been performed appropriately and rigorously? 

Reviewer #1: Yes

Reviewer #3: I Don't Know

4. Have the authors made all data underlying the findings in their manuscript fully available?

Reviewer #1: Yes

Reviewer #3: Yes

5. Is the manuscript presented in an intelligible fashion and written in standard English?

Reviewer #1: Yes

Reviewer #3: Yes

6. Review Comments to the Author

Reviewer #1: It was great to review this revised version. All comments were addressed and I do not have addition.

Reviewer #3: GENERAL COMMENTS:

Thank you for the opportunity to review this article. Generally the article addresses an important subject and is well conducted with important findings. Find below wome comments that might help improve the article.

TITLE:

It is not immediately clear what this study is about in the title, which is the evaluation of an intervention aimed at improving the household hungar and socioeconomic status of caretakers of ophans and vulnerable children. For any lay reader who is not familiar witth the term "WORTH Yetu", the topic is vague until further reading of the body of the article. Also, it is a good practice to state abbreviations that are not in general use and immediately obvious to any reader in full, before continueing to use the abbreviation. In the title, OVC is written as an abbreviation without any previous definition what it means.

ABSTRACT:

A minor edit required in the last sentence of the Abstract. "Recognising" has been written as "reconising"

BODY OF MANUSCRIPT AND CONCLUSION:

I have no particular concerns with the introduction, methods, results and conclusions of the study. The conclusion seem to be supported by the data presented.

7. PLOS authors have the option to publish the peer review history of their article (what does this mean?). If published, this will include your full peer review and any attached files.

Reviewer #1: No

Reviewer #3: No

---

## [Editor Report · Acceptance letter]

3 Apr 2024

PONE-D-23-25792R1 

PLOS ONE

Dear Dr. Exavery, 

I'm pleased to inform you that your manuscript has been deemed suitable for publication in PLOS ONE. Congratulations! Your manuscript is now being handed over to our production team.

Kind regards, 

on behalf of

Dr. Moses Katbi 

Academic Editor

PLOS ONE